# Selected Geoheritage Resources of "Atlantic Geopark" Project (Central Portugal)

**Salomé C. Custódio** [1,*] **, Maria Helena Henriques** [1] **, Emmaline M. Rosado-González** [2,3] **, Nuno M. Vaz** [2,3] **and Artur A. Sá** [2,3]

1   Geosciences Center, Department of Earth Sciences, University of Coimbra, Rua Sílvio Lima s/n, 3030-070 Coimbra, Portugal; hhenriq@dct.uc.pt
2   Department of Geology and Pole of the Geosciences Center, University of Trás-os-Montes e Alto Douro, Quinta de Prados, 5000-801 Vila Real, Portugal; emmalineg@utad.pt (E.M.R.-G.); nunovaz@utad.pt (N.M.V.); asa@utad.pt (A.A.S.)
3   UNESCO Chair on 'Geoparks, Sustainable Regional Development and Healthy Lifestyles', University of Trás-os-Montes e Alto Douro, Quinta de Prados, 5000-801 Vila Real, Portugal
*   Correspondence: scmcustodio@student.uc.pt

**Abstract:** The "Atlantic Geopark" Project corresponds to the first stage of a broad project addressing a future application to the Global Geopark Network of a territory located in Portugal: "The Atlantic Geopark: 600 million of geological history". It covers six central littoral and rural municipalities (Cantanhede, Figueira da Foz, Mealhada, Mira, Montemor-o-Velho, and Penacova), which display special and singular geodiversity, and it includes geological heritage with international relevance representing the opening and closing of the Rheic Ocean, the formation and breakup of Pangea, and the opening of the North Atlantic Ocean. Besides the geological heritage, here presented through the description and characterization of six geological sites (one per municipality) which served as anchors for the development of the project currently underway, the territory also provides other geoheritage resources related to uses of the local geological features. These resources hold significance in bolstering an application to the Global Geopark Network soon. They encompass partially artificial elements such as road excavations, agricultural soils, and quarries, as well as entirely artificial elements such as interpretation centers and museums. These elements serve as tangible representations of the various ways in which the Earth and local communities interact.

**Keywords:** UNESCO Global Geoparks; West Iberian Margin geological history; sites of geological interest; geo-itinerary; oceans route; geotourism; geoeducation

## 1. Introduction

Geological heritage can be defined as a selected part of Earth's geodiversity with scientific, educational, and/or tourism value which are worth preserving [1]. In the last decade, it has been proven that geoheritage plays an important role in socio-economic development in a territory, making it a valuable natural resource equivalent to mineral and/or hydrocarbon resources, among others. However, instead of being exploited for raw materials and/or energy production, geoheritage resources can be exploited for scientific, educational, and touristic purposes [2–4].

The geological heritage of a territory with international significance represents the starting point for any proposal aiming toward a future establishment of a UNESCO Global Geopark (UGGp), whose main pillars are focused on geoconservation, geoeducation, and geotourism [5]. Nevertheless, the mere geoheritage significance of a region does not suffice to substantiate a request for UNESCO recognition. Other requirements are needed, i.e., the territory must display a specific size and setting and must be managed by local people aiming at fostering economic development and education, as well as ensuring protection and conservation of its geological heritage [6]. Geological sites present in any UGGp

candidate hold significance as geoheritage assets when combined with other human-made structures found in the area, which emphasize the region's geodiversity and its connection with the local communities [7].

This work describes selected geological sites outcropping in six municipalities located in central littoral and rural of Portugal: Cantanhede, Figueira da Foz, Mealhada, Mira, Montemor-o-Velho, and Penacova. They represent some of the most emblematic milestones (at the study area) related to the closing of the Rheic Ocean and the breakup of Pangea [8] and the opening of the North Atlantic Ocean [9]. These geological sites hold the key to unravelling Earth's history in the territory, spanning more than 600 million years. These sites serve as a support to the ongoing project known as "The Atlantic Geopark", which seeks to gather evidence and pave the way for an application to the Global Geopark Network in the coming years. Furthermore, this study also documents additional geo-resources that are pertinent to such applications.

In addition to this, the territory needs a management structure fully committed with the local population's needs, which guarantees the physical integrity of its geo-heritage [6]. In this framework, the management structure behind this ongoing project is AD ELO—Local Development Association from the Bairrada and the Mondego—which oversees the management plan in the "planning stage" of the process.

Therefore, "The Atlantic Geopark" Project emerged from two baselines: its singular geodiversity and the recognition of local management structure is fully committed to the UGGp mission.

## 2. Geographical and Geological Setting

The research area is in the central coastal and rural regions of Portugal, encompassing six distinct municipalities (Cantanhede, Figueira da Foz, Mealhada, Mira, Montemor-o-Velho, and Penacova). The area sums up to 1452 km$^2$ and 60 km of the coastal line (from Mira to Figueira da Foz; Figure 1). This region is located half-way from both the capital of Portugal, Lisbon, and Oporto, making it very accessible to visitation with already several infrastructures built: a railway, a highway, and a seaport. In terms of demographic characterization, this territory, in 2021, had 162,686 residents [10], which makes a population density of 112 inhabitants/km$^2$.

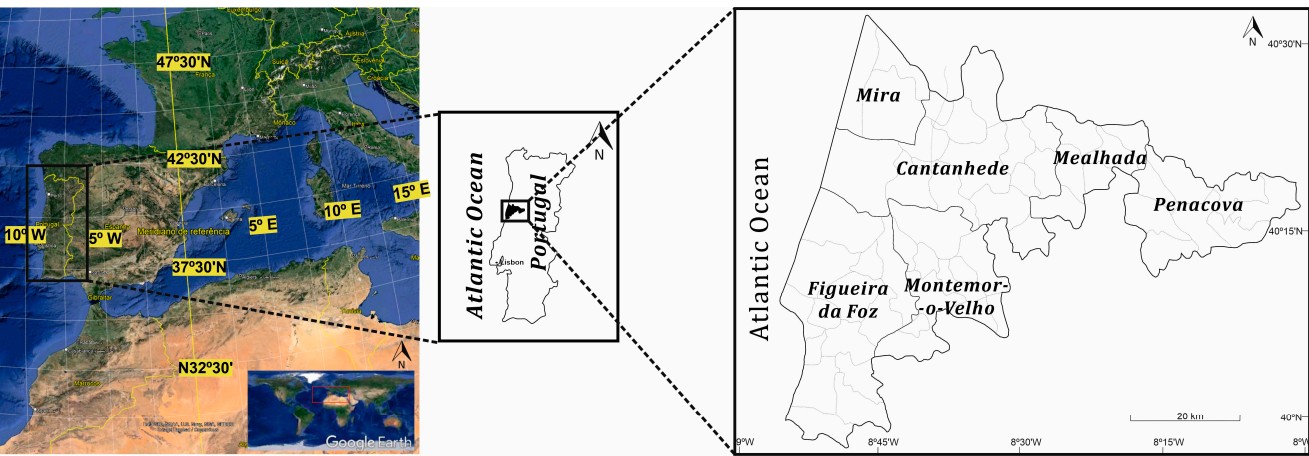

**Figure 1.** Location map of the study area (source of the cartographic data [11]; WGS84 geographic projection with an ESPG code of 4326).

From a morphostructural perspective, the study region includes predominantly the terranes of the Lusitanian Basin, a sedimentary basin of Meso-Cenozoic origin, extending towards the west. In contrast, the Iberian Massif occupies a smaller portion of the study area towards the east (Figure 2). The Porto–Tomar fault has brought together these two

distinct territories. The Porto–Tomar–Ferreira do Alentejo shear zone represents the intense deformation associated with this fault.

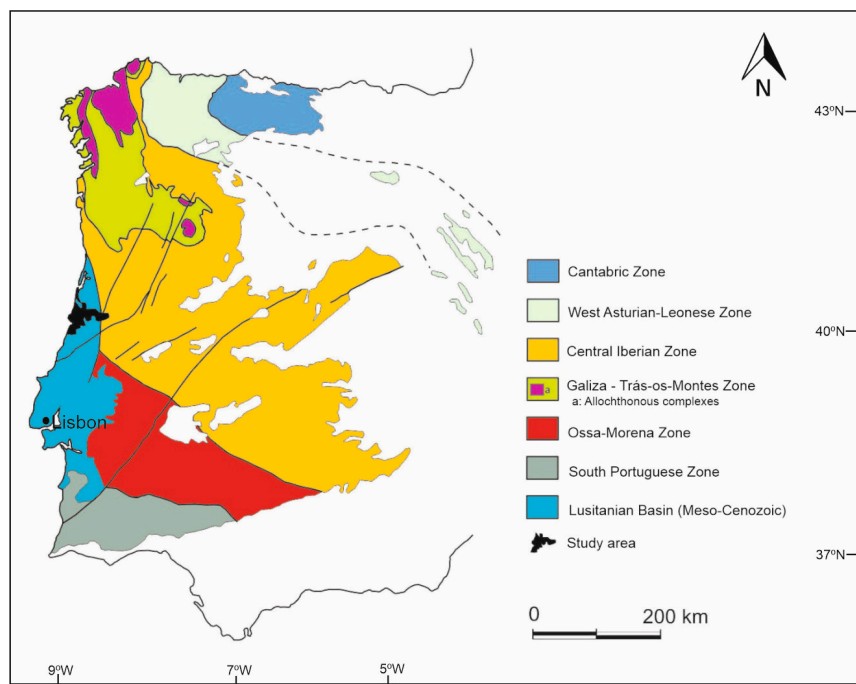

**Figure 2.** Tectonostratigraphic division of the terranes within the Iberian Peninsula with the location of the "Atlantic Geopark" Project territory (adapted from [12]).

*2.1. Iberian Massif*

The Iberian Massif is composed of several tectonostratigraphic terranes with specific characteristics, and, among them, there is the Iberian Terrane, which includes the Central Iberian Zone [13].

The Central Iberian Zone (CIZ) is limited to the north by the Vivero Fault, through which it contacts the West Asturian-Leonese Zone, and to the west by Porto–Tomar–Ferreira do Alentejo and Tomar–Badajoz–Cordoba shear zone, through which it contacts with the Ossa-Morena Zone. The CIZ includes two domains: the "Ollo de Sapo" domain and the "Schist-Greywacke Complex" domain [14–16], and the eastern part of the study area is integrated into the last domain. This consists of a metasedimentary sequence that extends from the Neoproterozoic to the Middle Cambrian (?), commonly referenced as the "Schist-Greywacke Complex", and currently differentiated into the Douro Group and the Beiras Supergroup [17]. The latter, characterized by non-carbonate turbiditic facies, outcrops in the territory and is assigned to the Neoproterozoic. Over this sequence, a metasedimentary sequence was deposited in angular unconformity, extending from the Ordovician to the Carboniferous [18]. The CIZ underwent substantial changes because of the Variscan Orogeny and the subsequent closure of the Rheic Ocean. These geological events played a crucial role in the assembly of the Pangea supercontinent.

*2.2. Lusitanian Basin*

The Lusitanian Basin occupies more than 20,000 km$^2$ in the Portuguese territory (onshore and offshore), extending along approximately 200 km of coast, in an NNW–SSE direction, and for more than 100 km in the perpendicular direction [19]. The Iberian Massif lies to the east, while the horst system, which is exposed in the Berlengas—Farilhões archipelago [20], can be found to the west. Towards the south, we have the Arrábida fault, and to the north, the Porto Basin [19] completes this configuration.

Its tectonic evolution was conditioned by the Mesozoic distension [21], related to the opening of the North Atlantic Ocean and the break-up of the Pangea supercontinent. The

stretching process began in the Late Triassic and its evolution lasted about 100 Ma [22]. The stratigraphy of the Lusitanian Basin comprises sediments dating from the Late Triassic to the end of the Cretaceous, reaching about 5 km in thickness, mostly from the Late Jurassic (about 3 km; [23]). The deposition of the sedimentary sequence in the Lusitanian Basin occurred on a rocky substrate from the Paleozoic, and an extensive exposure of this sequence can be observed in various areas of the basin [24]. It comprises four important intervals dated to the Upper Triassic–Lower Jurassic (Rift 1); Middle Jurassic–Upper Jurassic (SAG); Upper Jurassic–Lower Cretaceous (Rift 2 + Drift 1); and Lower Cretaceous–Cenozoic (Drift 2 + Inversion) [20,23,25], which influenced the basin geometry.

### 3. Geological Milestones Displayed at the "Atlantic Geopark" Project

In the "Atlantic Geopark" project territory, several milestones of Western Iberia geological evolution are well recorded, from the Neoproterozoic to Cenozoic (Figure 3). Some localities provide some of the most complete stratigraphic sequences of Portuguese geology, such as the Ordovician at the Buçaco Synclinal, the Jurassic at Cabo Mondego, and the Cretaceous at Salmanha.

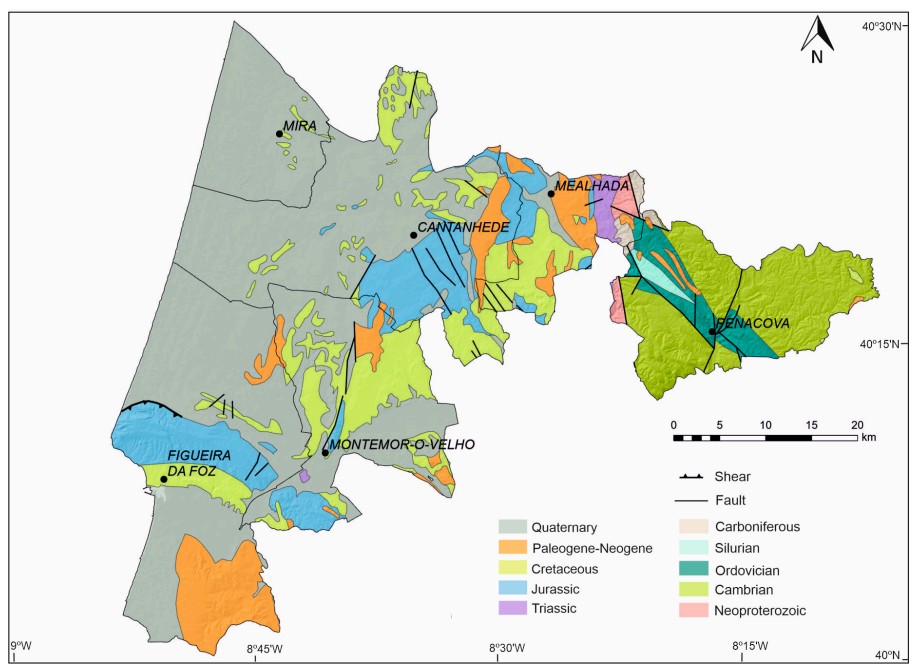

**Figure 3.** Geological map of the "Atlantic Geopark" Project territory (Original scale 1:500,000; project system: Datum Lisboa/Hayfor-Gauss EPSG: 5018; adapted from [9]).

The geological history of this territory starts east of the territory with the record of the Beiras Supergroup (635 Ma) [17] that represents the oldest record in the territory, and outcrops in Penacova area. This unit was deposited in the context of a back-arc sedimentary basin, associated with the evolution of a volcanic arc (Cadomian Orogeny; 650–550 Ma) which occurred on the margin of the Gondwana continent [26]. It was later inverted by the tectonic inversion associated with the opening of the Rheic Ocean, marking the beginning of the Variscan Cycle (Cambrian—Ordovician; 495–485 Ma), and another deformation episode related to the closing of the Rheic Ocean and the Pangea assembly [8].

The Buçaco Synclinal is a main structure that crops out in Penacova and Mealhada municipalities and lays out in angular disconformity with the Beiras Supergroup. This structure records an extraordinary well-preserved metasedimentary Paleozoic sequence spanning from the Lower Ordovician to the Silurian (485–423 Ma) [18], deposited in a passive margin context of the Rheic Ocean. It can attest to, e.g., the "Boda Event," a global event of climate change (453 Ma) [27] and a glaciation event (445 Ma) that led to the second major mass extinction [28] that marks the end of the Ordovician.

The Buçaco Carboniferous Basin outcrops in the Santa Cristina sector (Mealhada), representing the final stages of the Variscan Cycle [29]. Its units were deposited in a continental basin context located between the variscan mountains (Variscan Orogeny), and its infill is dated as Late Pennsylvanian (Early Ghzelian 303 Ma) [30].

The Silves Group, a siliciclastic rock succession dated from the Upper Triassic, also outcrops in the Mealhada municipality. This unit can confirm the central position that Iberia occupied in Pangea right before the opening of the North Atlantic Ocean, marking the break-up of that supercontinent. This siliciclastic sequence was deposited under semi-arid climate conditions in fluvial environments (alluvial fans and rivers), changing to shallow coastal depositional environments in early Jurassic times [31].

The Jurassic evolution of the North Atlantic Ocean is particularly well represented in Cantanhede and Figueira da Foz municipalities. The carbonate platform of the Proto-Atlantic is well recorded at Cantanhede outcrops. The Cape Mondego (Figueira da Foz) succession provides the most complete and continuous Jurassic record from Toarcian to Tithonian (184–149 Ma)—including the Bajocian GSSP (170 Ma) [32] and the Bathonian ASSP (168 Ma) [33]—and the syn-rift succession of the second rift of the Late Jurassic [34].

The marine transgression of the North Atlantic Ocean is well recorded in Montemor-o-Velho and Figueira da Foz, namely at the Baixo Mondego region. These outcrops record the transgressive/regressive cycles that represent the transition from late rifting to a passive margin, expressed by a large carbonate platform which extended far inland (Cenomanian—Turonian, 100–93 Ma) [35]. The paleontological heritage includes, besides representative macrofauna, the earliest angiosperms (Late Aptian to Early Albian; 121–113 Ma) [36].

The Quaternary geodynamics of the Atlantic Margin are well represented throughout the territory, with extensive superficial deposits that cover most of the littoral, dated from the Pleistocene to the Holocene [37,38].

## 4. Selected Geoheritage Resources of the "Atlantic Geopark" Project

Based on bibliographic research and detailed fieldwork, six geological sites (one per municipality) were selected with the following criteria: 1—to allow the creation of a geo-itinerary throughout the territory lasting one day and understandable to different types of audiences; 2—to integrate geological sites representative of the geological history of the territory, including the geological heritage of global or international significance, as required to establish any UGGp [5,6]; 3—to show different means of occupation and the use of its geological resources by local communities, therefore highlighting the links between geological heritage and all other aspects of the area's natural, cultural, and intangible heritages [5].

Selected geological sites are hereby presented and characterized in geochronological order and based on storytelling narrative techniques [39,40]. They support a geo-itinerary that make it possible to reconstruct the geological history of two oceans—Rheic and Atlantic: the Oceans Route (Figure 4).

### 4.1. "Livraria do Mondego": The Beginning of the Rheic Ocean (477–470 Ma)

The "Livraria do Mondego" (=Mondego's Bookshelf) at Penacova municipality (40°17′03″ N 8°15′50″ W) is a famous outcrop located on the right bank of Mondego River (the main river of this territory; Figure 5). The geological history of "Livraria do Mondego" started at approximately 477–470 Ma (Floian, Ordovician) and is represented by the Armorican Quartzite facies. This unit is part of a detrital sequence, essentially siliciclastic, which was formed in a supralittoral to infralittoral environment [41] in the context of the passive margin of the Rheic Ocean, at the northern border of the supercontinent Gondwana. In "Livraria do Mondego", the Armorican Quartzite was deformed by the compressive events that resulted from the closing of the Rheic Ocean and the collision of the continents that ultimately formed the Pangea supercontinent. The sediments underwent deformation during burial, resulting in their transformation into quartzite and their vertical alignment,

giving them the appearance of a bookshelf. This is why it is referred to as "Livraria do Mondego" in Portuguese.

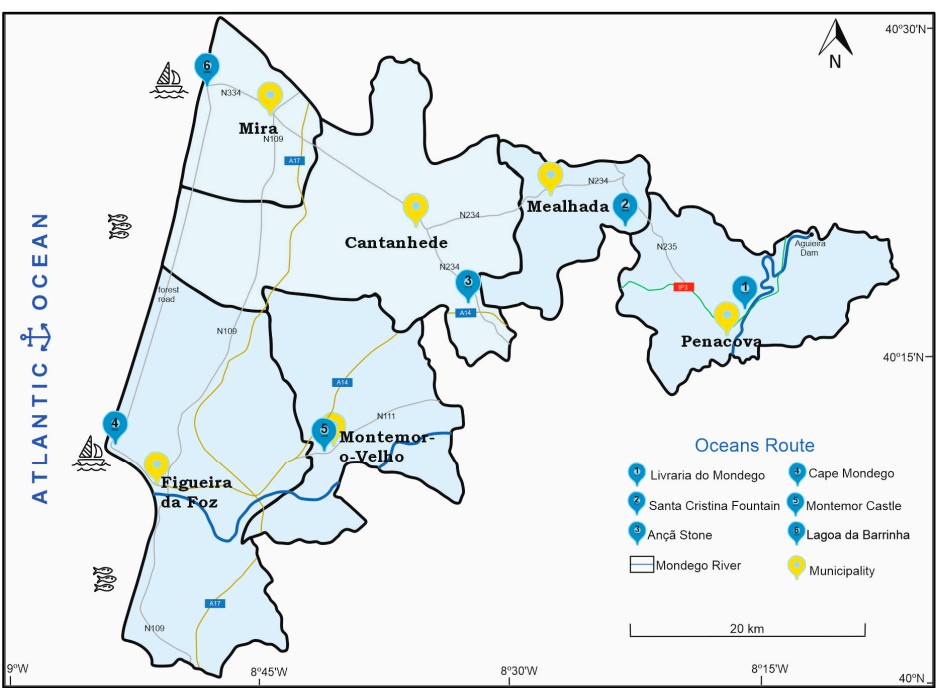

**Figure 4.** Location of the six geological sites of the "Atlantic Geopark" project territory referred in this proposal for the Oceans Route (source of the cartographic map: [11]; WGS84 geographic projection with an ESPG code of 4326).

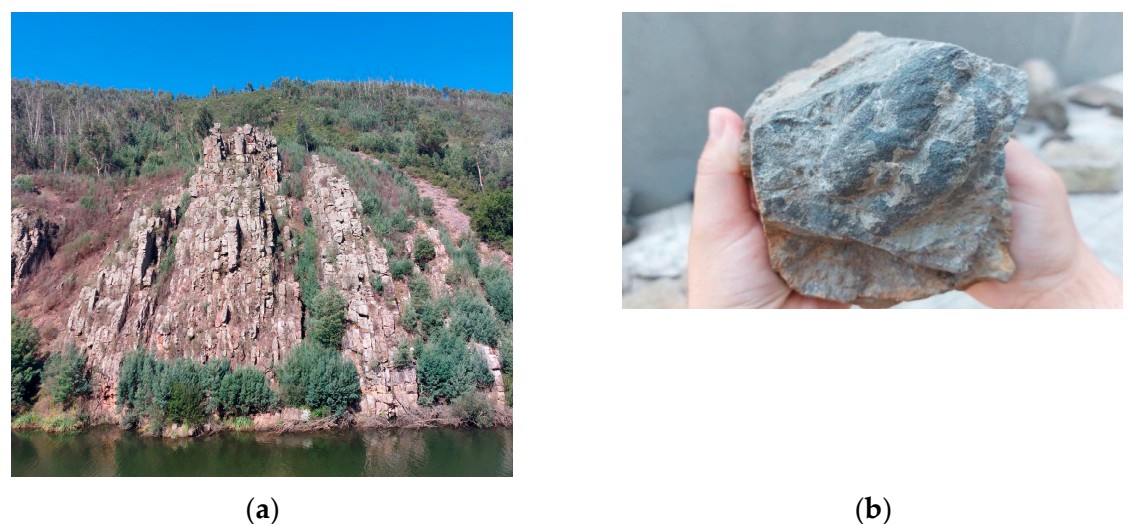

(**a**)                    (**b**)

**Figure 5.** (**a**) The geological site of Livraria do Mondego, where is possible to follow multiple tectonic processes: 1. The burial and metamorphization of the sediments; 2. The uplifting resulted from the compressive forces, verticalizing the layers; 3. The fracturing on the quartzites, allowing the carving the Mondego River.; (**b**) resting traces of *Rusophycus*.

It is possible to enjoy the scenic content sensu [42] of this geological site from a viewpoint, or through down-river activities, whether through extreme sports or through more leisurely options like simply take a stroll on the "Barca Serrana" ("Mountain Boat"). For centuries, "Barca Serrana" was the main form of transportation and communication for

the population around the Mondego River, a tradition that is now being restored by the local population for tourist purposes to preserve their cultural heritage.

*4.2. Santa Cristina Foutain: Nestled Amidst Towering Mountains (315–303 Ma)*

The Santa Cristina Fountain at Mealhada municipality (40°20′55.1″ N 8°23′02.6″ W corresponds to an important outcrop of the Monsarros Formation (lower Gzhelian, Pennsylvanian, Carboniferous; [30]), one of the three units of the sedimentary sequence that composes the synclinal structure of the Buçaco Carboniferous Basin. The last one is a continental intramountainous basin that was formed due to the compressive movements related to the Variscan Orogeny, marking the end of this cycle of Wilson. The geological record of Santa Cristina presents particularities from a paleoclimatic and paleoenvironmental point of view, namely the disappearance of typical floras of humid climates and the appearance of others adapted to dry climates (*Cordaites*—trunk and roots, *Calamites*) ([29]; Figure 6). This geological site shows current horsetail plants (*Equisetum hyemale*) and some specimens of their oldest ancestors, corresponding to arborescent horsetails (*Calamites* spp.).

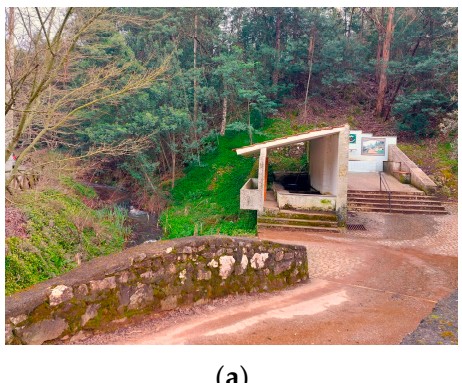
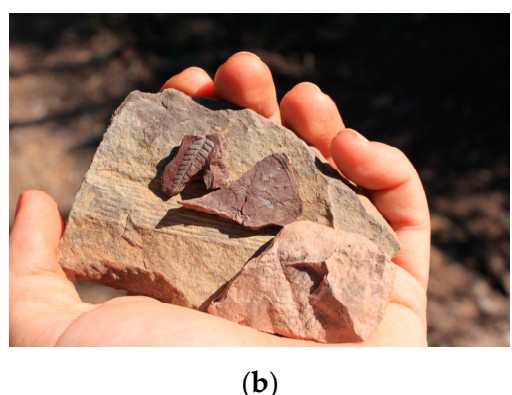

(**a**)　　　　　　　　　　　　　　　　　(**b**)

**Figure 6.** (**a**) Santa Cristina Fountain; (**b**) *Cordaites* and *Pecopteris*.

*4.3. "Pedra de Ançã": The Carbonate Platform of the Proto-North Atlantic (170 Ma)*

The "Ançã Stone" is a micritic limestone belonging to the Ançã Formation (Bajocian—Bathonian) that outcrops in the Cantanhede municipality (40°17′18.1″ N 8°32′22.4″ W), and it is associated with the development and extension of the proto-North Atlantic, representing the previous stages of the North Atlantic's opening [9,43,44]. The Ançã Formation was deposited in the context of a carbonate ramp with high fossiliferous content, known particularly for its record of well-preserved ammonites (Figure 7).

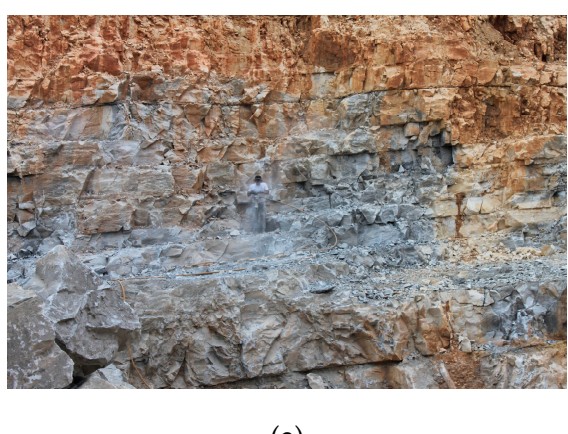
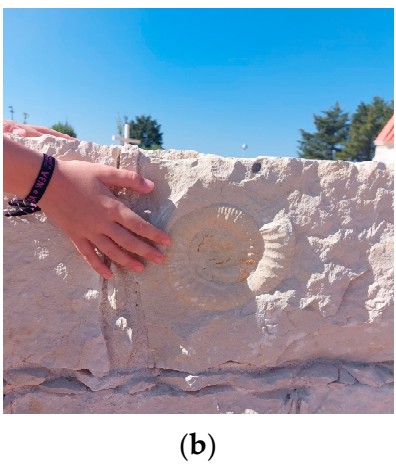

(**a**)　　　　　　　　　　　　　　　　　(**b**)

**Figure 7.** (**a**) Ançã limestone at Boiça quarry; (**b**) detail of *Ammonite* on an urban wall.

The beginning of the use of Ançã limestone dates back to the 14th century [45]. Its "malleable" characteristic allowed the architects and sculptors to build monasteries, churches, chapels, buildings, and statues that played an important role in artistic movements and architecture styles in Portugal and overseas. Pieces of "Ançã stone" can be found scattered all around the globe, especially in Santiago de Compostela, Brazil, Japan, Goa, Guinea Bissau, Angola, Macao, Jerusalem, and Rome [46].

The limestones from the Ançã Formation are still explored in multiple active quarries; however, the "Ançã Stone" is no longer available to explore. In that context, the "Museu da Pedra" (=Stone Museum) in the Cantanhede municipality was built to preserve and honor the ancient practices of the exploration of the stone as well as the stonework made by the sculptors.

### 4.4. Cape Mondego: The Opening of the North Atlantic Ocean (184–149 Ma)

Classified as Natural Monument in 2007, the Cape Mondego at Figueira da Foz municipality (40°10′49.6″ N 8°54′20.9″ W) represents the most continuous Jurassic outcrop in Europe with global relevance ([47]; Figure 8). In this geological site, it is possible to follow the different stages of the opening of the North Atlantic Ocean.

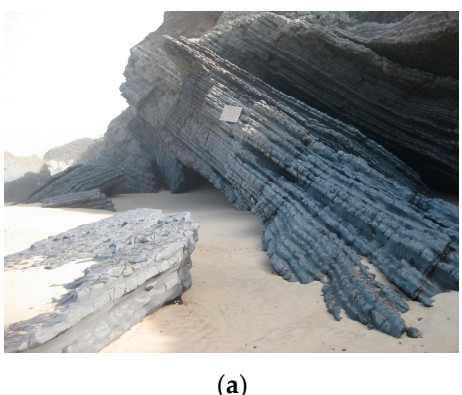 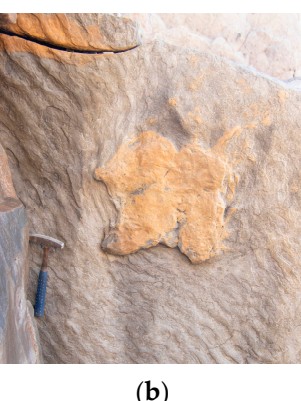

(**a**)        (**b**)

**Figure 8.** (**a**) GSSP at Murtinheira section in Cape Mondego; (**b**) inverted dinosaur footprint.

Starting from Cape Mondego north (Middle Jurassic), the record shows a monotonous sedimentation where the ammonites were thriving in that environment. This rich record, along with other biostratigraphic tools, allowed the experts to establish the Global Standard Stratotype and Point (GSSP) for the Aalenian–Bajocian boundary [32]. The second-best outcrop for the Bajocian—Bathonian age (ASSP—Auxiliary Standard Stratotype and Point) is also in Cape Mondego [33]. This outcrop is commonly known for its morphology as a "Porta-aviões" ("aircraft carrier"), especially by local fishermen.

Continually to the south, the middle Oxfordian–Tithonian (Upper Jurassic) section provides an excellent continuous sedimentary record of the evolution of the second rifting episode of the Lusitanian Basin [34]. This section presents the initial phase of the rift climax, characterized by the occurrence of siliciclastic units. It then transitions into the middle phase, which is marked by a transgressive stage consisting solely of carbonate facies association. Finally, the late phase of the rift climax is depicted by a thick prograding siliciclastic unit, primarily of deltaic origin, overlain by additional terrigenous deposits from the Cretaceous period [34].

The Cape Mondego represents not only the geoheritage of stratigraphic and paleontological types, but also other values related to mining and industrial heritage. Cape Mondego was the first coal mine in Portugal (1773; [48]), whose activity resulted in the publication of the first research study about dinosaur footprints in Portugal, discovered in 1884 [49], that currently integrates the collection at the National Museum of Natural History and Science in Lisbon.

### 4.5. Castle of Montemor-o-Velho Viewpoint: A Cultural Landscape (1.80 Ma)

From the castle viewpoint in Montemor-o-Velho (40°10′32.6″ N 8°40′58.1″ W), it is possible to enjoy a panoramic view of the "Baixo Mondego" landscape that marks the southern part of the territory. This landscape is conditioned by the alluvial plains of the Mondego River and is characterized by very extensive rice and corn fields. "Taipal" is the name given to the marsh located in these alluvial plains, classified as a Ramsar Wetland Site since 2001 (Ramsar site no. 1107; [50]), which is an important area for the biodiversity of the region and contributes to the local regulation of the underground water (Figure 9).

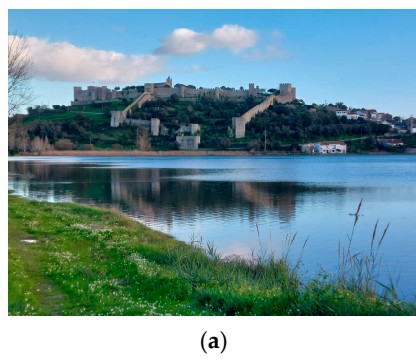
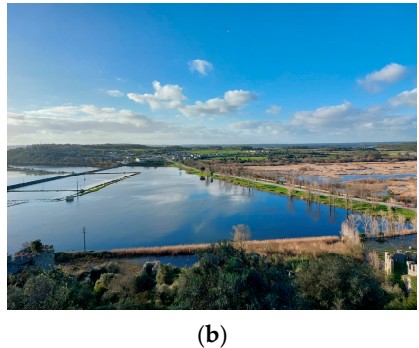

(**a**)  (**b**)

**Figure 9.** (**a**) Castle of Montemor-o-Velho; (**b**) rice fields and the Taipal marsh.

In turn, the castle itself played a key role in the Christian Reconquest, being the main Fortress of the Baixo Mondego defense system in medieval times [51]. This military and historical feature attracts a specific type of tourist and promotes "Militar Tourism", through routes particularly developed for those enthusiastic about military history [52].

### 4.6. "Lagoa da Barrinha": The Current Atlantic Margin (0.0056 Ma)

Extensive dune fields (Holocene; [37]) and freshwater lagoons, considered a singular expression in the national territory [53], mark the northern part of the territory, called the "Gândara" region.

The Barrinha lagoon in Mira (40°27′09.7″ N 8°48′04.2″ W) resulted from a geomorphological feature called "Ria de Aveiro", the only delta system in Portugal. The meridional sector of this haff-delta is in Mira, and it was formerly connected to the Atlantic. Currently reduced to a lagoon, the "Barrinha de Mira" is now linked to the Ria de Aveiro through an artificial channel (the Mira channel), and it is mainly used for recreational purposes such as nautical sports, walks, and bike trails (Figure 10a).

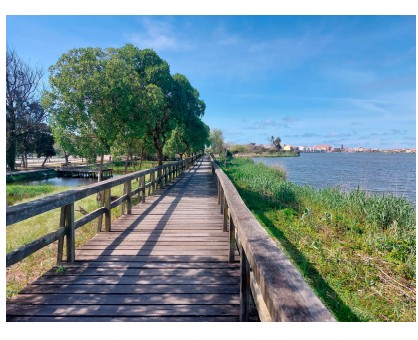
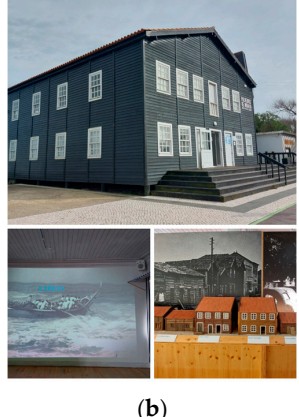

(**a**)  (**b**)

**Figure 10.** (**a**) Perspective of the Barrinha lagoon; (**b**) Mira Ethnographic museum.

Ancient practices directly related to the ocean marks this part of the territory, and it can be revived though the Mira Ethnographic Museum, right next to the Barrinha Lagoon. Here,

they explore the former "Arte Xávega" (artisanal fishing), or they show how they preserved the "Palheiros de Mira" (stilt buildings) as a part of the cultural heritage (Figure 10b).

## 5. A Geo-Itinerary for All

This region showcases various types of geoheritage across different scales of significance and with diverse applications, which can be observed by following the suggested geo-itinerary that can be comfortably completed within a single day (Table 1). The six geological sites recommended in this context fulfil these criteria and can be initiated from any of the six municipalities encompassed within the project.

**Table 1.** Characterization of each geological site of this work based on its geoheritage type, potential use, relevance scale, and typology.

| Geological Sites | Geoheritage Type | Potential of Use | Relevance Scale | Typology |
|---|---|---|---|---|
| "Livraria do Mondego" | Paleontological Stratigraphical Tectonic Geomorphological Sedimentological | Educational Touristic | National | Area |
| Santa Cristina Fountain | Paleontological Stratigraphical Hydrogeological | Scientific Educational Touristic | Local | Area |
| "Pedra de Ança" | Paleontological Stratigraphical | Educational | National | Point |
| Cape Mondego | Paleontological Stratigraphic Sedimentological Geomorphological | Scientific Educational Touristic | Global | Complex Area |
| Castle of de Montemor-o-Velho | Geomorphological | Educational Touristic | Local | Viewpoint |
| "Lagoa da Barrinha" | Geomorphological | Touristic | Local | Point |

Considering that the geoscience themes in Portugal appear in curriculum documents [54] from the first grade (from six to ten years old) and are present all throughout the Portuguese compulsory education (from six to eighteen years old), this prepares all citizens for a family visit or other types of visits tobe sensible about the geoscience themes which are represented at the six geological sites chosen for the Oceans Route (Table 2).

**Table 2.** Examples of different curriculum questions and skills that students should develop in each school year according to Essential Learning (Portuguese education curriculum; [54]), related to geoscience themes at the selected geological sites.

| Geological Sites | Issues/Abilities | Grades (Ages) |
|---|---|---|
| "Livraria do Mondego" | Land in transformation - Relate the expansion and destruction of the ocean floor to the theory of plate tectonics (boundaries between plates) and the constancy of the Earth's volume and mass | 7th (12–13) |
| Santa Cristina Fountain | Sedimentation and sedimentary rocks - Explain the importance of fossils (age/facies) in relative dating and the reconstitution of palaeoenvironments | 11th (16–17) |

**Table 2.** *Cont.*

| Geological Sites | Issues/Abilities | Grades (Ages) |
|---|---|---|
| "Pedra de Ançã" | Land in transformation<br>- Explain processes involved in the formation of sedimentary rocks (sedimentogenesis and diagenesis) presented in different media (schemes, figures, and texts) | 7th<br>(12–13) |
| Cape Mondego | Geology and methods<br>- Use principles of geological reasoning (actualism, catastrophism, and uniformitarianism) in interpreting evidence of Earth-history facts (stratigraphic sequences, fossils, rock types, and landforms) | 10th<br>(15–16) |
| Castle of de Montemor-o-Velho | Society/nature/technology<br>- Recognize and evaluate the natural and cultural heritage—local, national, etc.—and identify the natural elements in a landscape (e.g., geosites) | 4th<br>(9–10) |
| "Lagoa da Barrinha" | Water, air, rocks, and soil—terrestrial materials<br>- Interpret diverse information about the availability and circulation of water on Earth, enhancing your knowledge from other disciplines (e.g., history and geography of Portugal). | 4th<br>(9–10) |

## 6. Discussion

The "Oceans Route" here proposed can contribute actively to promoting geoeducation, as required for any UGGp [6,55], and represents the first initiative of its kind at this territory. Learning about the geological history of the Western Iberian Margin in a one-day journey is no easy task. However, numerous educational institutions only allocate a limited amount of time for a field trip, requiring the need for a diverse range of curricular and extra-curricular activities that address various academic concerns and cater to the different skill levels of the school community [56–59].

Geoconservation is another requirement for a territory to become a UGGp [55], and several initiatives have already been implemented within the region concerning geoconservation. Indeed, both national and local governments recognize the preservation and appreciation of its geological heritage. This is evident in the case of the Cabo Mondego geological site, which has been officially designated a Natural Monument since 2007 (Regulatory Decree n. ° 82/2007, 3 October). Similarly, the Ançã limestone is currently undergoing the process of being recognized as a Global Heritage Stone Resource within the framework of the IGCP Project 637 [60]. In addition, concerning geoheritage inventories, both areas covered by this study (Iberian Massif—Central Iberian Zone and the Lusitanian Basin) are included in geological contexts of international relevance [47,61], and geological site 4 (Cape Mondego) is included in two different data bases of geological interest sites in Portugal—one managed by the Nacional Laboratory of Energy and Geology (LNEG) [62] and the other by the ProGeo Iniciative [63]. Concerning the natural protected areas at the territory, the European initiative, Rede Natura 2000, also includes [64]: site PTCON005—"Dunas de Mira, Gândara e Gafanhas"; site PTZPE0060—"Aveiro/Nazaré" (offshore); PTCON0063—"Maceda/Praia da Vieira" (offshore); Arzila Wetland (Ramsar site no. 822), and Taipal Wetland (Ramsar site no. 1107), all managed, like the Cape Mondego Natural Monument, by the Institute for Nature Conservation and Forests (ICNF).

The territory has a truly diverse tourism offer, from long white-sand beaches at the coast to inland reference spas [65]. The region boasts fortresses, churches, and diverse architectural styles that highlight its cultural heritage. Its gastronomic offerings cater to all tastes, while its wines offer unique flavors. The traditions and customs of the local communities add richness to the tourism experience. To enhance tourism, geotourism proposals can be introduced (such as Heritage Routes [66]), highlighting the region's natural wonders and geological features. This would attract visitors year-round, providing an unforgettable experience.

Following the UNESCO requirements of self-evaluation [55], a first preliminary assessment of the six selected geoheritage resources here presented support the idea that the territory of the "Atlantic Geopark" project displays the required features and contributes actively to the UGGp mission: geoconservation, geoeducation, and geotourism.

## 7. Conclusions

This work identifies and characterizes some important features of the project "Atlantic Geopark" currently underway, addressing a future application to the Global Geopark Network. It involves six municipalities of central littoral and rural Portugal (Cantanhede, Figueira da Foz, Mealhada, Mira, Montemor-o-Velho, and Penacova) in a total area of 1452 km$^2$ and 60 km of coastal line. Six geological sites were selected (one per municipality), aiming at tellingthe geological history of the Western Iberian Margin since the Early Paleozoic, which includes the closing of the Rheic Ocean, the breakup of Pangea, and the opening of the North Atlantic Ocean. The itinerary provided in this document—the Oceans Route—offers a comprehensive overview of the geological events that have occurred in the past 600 million years within the specified territory. Additionally, it showcases the diverse range of geological features found in the area, including global significant geological heritage sites. Furthermore, it emphasizes the close connection between the natural and cultural aspects of the region, making it an ideal resource for educational purposes and attracting tourists.

Based on the aforementioned circumstances, it is evident that all the prerequisites are fulfilled for this region to undertake and execute an application aimed at seeking recognition as a UNESCO Global Geopark.

**Author Contributions:** Conceptualization, S.C.C. and M.H.H.; methodology, S.C.C., M.H.H. and A.A.S.; validation and investigation, M.H.H., N.M.V., E.M.R.-G. and A.A.S.; writing—original draft preparation, S.C.C. and M.H.H.; writing—review and editing, S.C.C., M.H.H., E.M.R.-G., N.M.V. and A.A.S.; supervision, M.H.H. and A.A.S.; project administration, N.M.V. All authors have read and agreed to the published version of the manuscript.

**Funding:** This study was supported by (i) University of Trás-os-Montes e Alto Douro/Geosciences Center with a Research Grant [BI/UTAD/96/2021], (ii) Portuguese funds from Fundação para a Ciência e a Tecnologia, I.P. (Portugal) in the frame of UIDB/00073/2020 (https://doi.org/10.54499/UIDB/00073/2020; accessed on 12 March 2024) and the UIDP/00073/2020 (https://doi.org/10.54499/UIDP/00073/2020; accessed on 12 March 2024) projects of the I & D unit of Geosciences Center (CGEO). This research is a contribution of the UNESCO Chair in "Geoparks, Sustainable Regional Development and Healthy Lifestyles" of the University of Trás-os-Montes e Alto Douro (Portugal), for the activities of the Portuguese National Committee for the International Geosciences Program of UNESCO (IGCP).

**Acknowledgments:** The authors are grateful to Patrícia João for the discussion concerning curricular issues/abilities, and to Gustavo Gonçalves Garcia for all the help with the figures. The authors also thank the two anonymous reviewers for their comments and suggestions.

**Conflicts of Interest:** The authors declare no conflicts of interest.

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
