# Peer review of "Selected Geoheritage Resources of “Atlantic Geopark” Project (Central Portugal)"

_geosciences, doi:10.3390/geosciences14030081_

Round 1

Reviewer 1 Report

Comments and Suggestions for Authors

This article shows six locations that serve as a common thread for geo-educational and geo-touristic activities, and that value the geo-conservation of an area in Portugal so that in the future it can be declared a Geopark by UNESCO.

The work is interesting and includes the most important elements that this type of proposal should have, and I do not find any problems in its drafting, so I recommend that it be accepted with minor corrections. 

- From a geo-educational point of view, I think it would enrich the text to specify which educative levels could be targeted for visits to these areas and what prior knowledge would be desirable for people who would like to follow this geological route.

-Below are some suggestions and some minor mistakes that should be corrected:

Line 46: Put a full stop between "[6]" and "Geological sites".

Line 54: Please add at least one reference for each of these geotectonic events or processes named in this sentence.

Line 200: In the geographical coordinates replace "O" with "W".

Line 220: In image (a), in addition to naming the area, it would be desirable to add a sentence explaining what can be seen, even if it is also being commented on in previous paragraphs.

Line 236: image (a) is not sufficiently clear or explanatory. If possible, it would be desirable to replace it with a better quality image or try to focus this image on the most important aspects described in the text.

Line 242: remove extra space between the full stop and "The Ança Formation...".

Line 260: In the geographical coordinates replace "O" with "W".

Line 282: add a space between "1884" and "[49]".

Line 304: the reference "Martins et al., 2006" does not comply with the journal's guidelines.

Line 318: if possible, it would be good to provide a picture of the interior showing these named exhibits.

Line 334: Delete the full stop after [54] or replace it with a comma.

Comments on the Quality of English Language

English style and grammar are correct.

Reviewer 2 Report

Comments and Suggestions for Authors

Dear authors,

The part dedicated to the tecto-stratigraphic characterisation is quite good, summarised and to the point, in order to contextualise the study area, and which will be the basis for understanding the geological milestones of the geopark project. 

I find very interesting the selection of the geological milestones that represent the geology of the study area, the geopark project, and the chronological and "informative" discourse: Rheic and Atlantic: the oceans route. Also, the brief and concise characterisation of each of the selected geosites, one per municipality, which argue this common thread of the geology of the territory.

Are there any other geological landmarks with some kind of protection as natural protected areas that have not been selected as "geological sites"? I understand that they are included in some kind of inventory of geological interest sites of international relevance carried out by the "geological service" of Portugal, if so, why is this not indicated? (such as Ramsar Wetland Baixo Mondego - geosite: Castle of Montemor-o-Velho).

Are the tecto-stratigraphic areas in which the study area is included included as "geological contexts of international relevance"? Although it is mentioned in the objectives, that the basis for the selection is their international relevance .... (lines 182-184), and for two of the selected geosites (Cape Mondego, lines 265-266); Ança Limestone as a Global Geoheritage Stone Resource, but not for all of them.

Are any of these landmarks part of the international inventory of the Global Geosites programme? If not, you should discuss on what geoheritage is protected in the study area, and on what remains to be protected and become, not only landmarks of the Geopark project, but also potential sites of interest in the Global Geosites programme.

Other comments and suggestions to improve the final quality of the manuscript for publication:

-Line 28-29: the names of the municipalities should be replaced by other more specific keywords of geoheritage, geology, geotourism in the area, or even Portugal.

-Lines 78-79: The authors present a location map (Figure 1) of the study area, but it would be advisable to include the geographical coordinates, projection system, source of the cartographic data, authorship of the map. In addition, in the caption of the photo, you state that it is a "geographical setting", so you should include more information if this is the case. It is suggested that you indicate that it is a "location map of the study area". 

-Line 87 (Figure 2) and 175 (Figure 3): It would be advisable that you also include coordinates, authorship of the map, source of cartographic data, etc., especially on the map in Figure 3, which corresponds to the study area and needs to be "located" with its coordinates. 

-Lines 193-196: Is it a geotourism map? It looks like a geotourism map given the indications and symbology you have used. You talk about "geological sites", and I think that's fine, but wouldn't it be more appropriate to write "geosites", which is the most internationally used concept for UNESCO global geoparks? The map, in addition to incorporating the basic elements indicated in the previous figures (coordinates, authorship, source of the cartographic base...), should include basic geographic information such as population centres, roads, rivers, etc., to serve as a "guide" to know where the "geo-itinerary" runs. And finally, the numbers that appear on the map corresponding to each of the selected geological sites should be indicated (indicate the name of each one of them on the legend or in the caption). 

-Line 304: change the citation (Martins et al., 2006) by the corresponding number, according to the order of appearance, and re-order the numbering of citations if this change affects the rest of the citations. 

-Line 319: This section goes directly to the different values (use, relevance, type...) of the geoheritage represented in the selected geosites. It would be more appropriate for this section to be separate from the description of the selected geosites, or, alternatively, within section 5 "discussion". A more extensive discussion of table 1 is also missing.

-Lines 327-352: In addition to incorporating the discussion in table 1, you should reflect on the international relevance of the geological contexts represented by the 6 geological sites selected by the authors. Although you indicate that 3 are of local relevance, 2 national and only 1 global, any proposal for a UGGP must be accompanied by the relevance that this territory contributes to international geology, in order to present the geopark candidacy dossier to both the national geoparks committee (or the Portuguese institution in charge of these matters) and the international UNESCO global geoparks committee. Therefore, is there any justification you can provide for the selection of these particular geosites.

As for the protected natural areas, you should reflect on the geoheritage that is protected in the study area (Cape Mondego as Natural Monument and Baixo Mondego as Ramsar Wetland) and on what remains to be protected according to the geoconservation precepts of the UNESCO Global Geoparks, not only in landmarks of the geopark project, but also in possible sites of interest in the Global Geosites programme (including the proposal of Ança Limestone as a Global Geoheritage Stone Resource). 

On the other hand, is there protection of the historical-cultural heritage linked to the geoheritage of the territory? For example, world heritage sites, or local or national protection. Not mentioned.

Furthermore, the presentation of any candidacy of a UGGP implies that the territory is functioning as if it were already a geopark in its own right (even if it is not recognised by UNESCO), and not only from the point of view of the selection of geosites of interest. In this sense, I miss the mention of whether environmental education activities (geo-education) are being carried out on the basis of the proposed geo-itinerary (has it been implemented yet? ); whether there are training activities for local guides in geoheritage and geotourism; whether there are activities linked to geotourism itself (I am not talking about sun and beach tourism); whether there is a link between the gastronomic offer and geoheritage within geotourism activities; or whether there are dissemination, scientific, artistic, etc. activities related to geoheritage of the territory and the proposed candidacy as a Unesco Global Geopark, as has been done in other candidate territories for UNESCO Geopark, before submitting their candidacy (e.g. Calatrava Volcanoes. Ciudad Real Geopark Project in Central Spain; Serra da Estrela Geopark, Portugal, among others).

-Lines 353-370, Section 6 Conclusions: This section is very similar to the abstract. Perhaps you should make a final consideration on the needs of the geopark project that have not yet been made on the basis of the UNESCO nomination requirements: a wider selection of geosites of international relevance, extension of the number of protected sites in the study area, the need for specific geo-education and geotourism activities, if any... you should even reflect on what the declaration of a geopark in this territory would bring, both from the point of view of geoconservation and also for the socio-economic development of the territory.

-Figures 5, 6, 7, 8, 9 and 10: Provided that the editorial rules allow it, the images could be enlarged so that they can be better seen in the framing of the manuscript. 

-Figure 6 a (line 235): Is there another image of Santa Cristina Fountain of better quality and more detail?

I hope that my suggestions will improve the quality and relevance of the manuscript for further publication.

Best regards.

Round 2

Reviewer 2 Report

Comments and Suggestions for Authors

Dear authors,

I understand your arguments in relation to the suggestions for change that were proposed in the review, mainly to increase the relevance of the work for future UNESCO geopark projects that want to take it as an example.

The rebuttal arguments that make up the review are precisely what should have been included in the manuscript, especially in the discussion and conclusions. I am referring to what is related to the global inventory of geological contexts of international relevance that the Portual geological service has listed, the protected natural spaces present in the territory (apart from those mentioned), educational, geotourism, training activities that exist, or no, linked to the geoheritage of the territory and its relevance.

I know that several of the suggestions are "not the objective of the work", as they argue in their reply, of course, but the suggestions were made to improve the quality and impact of the manuscript internationally (scientific soundness) and as a reference for other future geopark projects, as I told you.

With all this, I do not mean that the work is of low quality, on the contrary.

Last assessment: In the manuscript document that I have received, the map in figure 1 (lines 78-79), the geographical coordinates are not clearly visible, it is suggested that you make the coordinate size larger.

Kind regards.

Author Response

Thank you very much for your concern.

We understand your point of view, so we added information regarding the “the global inventory of geological contexts of international relevance that the Portugal geological service has listed” and “the protected natural spaces present in the territory (apart from those mentioned before)”.

In terms of “educational, geotourism, training activities that exist, or no, linked to the geoheritage of the territory and its relevance”, there are few or none of the kind. This is really the first stage of the project. And we believe that this “Ocean route” represents the first initiative of geoeducation at the study area as a whole.

Figure 1 has been improved according to your suggestion.